genomics, evolution, palaeontology

ancient DNA, Battle Axe Culture, Corded Ware Culture, demography, European Neolithic, migration

**Authors for correspondence:**
Helena Malmström
e-mail: helena.malmstrom@ebc.uu.se
Torsten Günther
e-mail: torsten.guenther@ebc.uu.se
Mattias Jakobsson
e-mail: mattias.jakobsson@ebc.uu.se

[†]These authors contributed equally to this study.

### PUBLISHING

# The genomic ancestry of the Scandinavian Battle Axe Culture people and their relation to the broader Corded Ware horizon

Helena Malmström[1,2,†], Torsten Günther[1,†], Emma M. Svensson[1], Anna Juras[3], Magdalena Fraser[1,4], Arielle R. Munters[1], Łukasz Pospieszny[5,6], Mari Tõrv[7], Jonathan Lindström[8], Anders Götherström[9], Jan Storå[10] and Mattias Jakobsson[1,2]

[1]Human Evolution, Department of Organismal Biology, Uppsala University, 752 36 Uppsala, Sweden
[2]Centre for Anthropological Research, Department of Anthropology and Development Studies, University of Johannesburg, 2006 Auckland Park, South Africa
[3]Department of Human Evolutionary Biology, Institute of Anthropology, Faculty of Biology, Adam Mickiewicz University in Poznań, ul. Uniwersytetu Poznańskiego 6, 61-614 Poznań, Poland
[4]Department of Archaeology and Ancient History, Uppsala University–Campus Gotland, 621 67 Visby, Sweden
[5]Department of Anthropology and Archaeology, University of Bristol, Bristol BS8 1UU, UK
[6]Institute of Archaeology and Ethnology, Centre for Studies into Late Antiquity and Early Medieval Times, Polish Academy of Sciences, 61-612 Poznań, Poland
[7]Department of Archaeology, Institute of History and Archaeology, University of Tartu, 50090 Tartu, Estonia
[8]Graduate School of Contract Archaeology, Department of Archaeology, Linneaus University, 391 82 Kalmar, Sweden
[9]Archaeological Research Laboratory, Department of Archaeology and Classical Studies, and [10]Osteoarchaeological Research Laboratory, Department of Archaeology, and Classical Studies, Stockholm University, 106 91 Stockholm, Sweden

HM, 0000-0002-6456-8055; TG, 0000-0001-9460-390X; MJ, 0000-0001-7840-7853

The Neolithic period is characterized by major cultural transformations and human migrations, with lasting effects across Europe. To understand the population dynamics in Neolithic Scandinavia and the Baltic Sea area, we investigate the genomes of individuals associated with the Battle Axe Culture (BAC), a Middle Neolithic complex in Scandinavia resembling the continental Corded Ware Culture (CWC). We sequenced 11 individuals (dated to 3330–1665 calibrated before common era (cal BCE)) from modern-day Sweden, Estonia, and Poland to 0.26–3.24× coverage. Three of the individuals were from CWC contexts and two from the central-Swedish BAC burial 'Bergsgraven'. By analysing these genomes together with the previously published data, we show that the BAC represents a group different from other Neolithic populations in Scandinavia, revealing stratification among cultural groups. Similar to continental CWC, the BAC-associated individuals display ancestry from the Pontic–Caspian steppe herders, as well as smaller components originating from hunter–gatherers and Early Neolithic farmers. Thus, the steppe ancestry seen in these Scandinavian BAC individuals can be explained only by migration into Scandinavia. Furthermore, we highlight the reuse of megalithic tombs of the earlier Funnel Beaker Culture (FBC) by people related to BAC. The BAC groups likely mixed with resident middle Neolithic farmers (e.g. FBC) without substantial contributions from Neolithic foragers.

## 1. Introduction

An influential wave of migration into central Europe from the Pontic–Caspian steppe occurred *ca* 3000 before common era (BCE) [1,2]. The Yamnaya expansion brought new genetic ancestry to Neolithic Europe, which still today is a

part of the genetic variation [3]. Northwest of the Yamnaya complex, we find the Corded Ware Culture (CWC) complex that was distributed over central and northern Europe between 3000/2800 and 2300/2000 BCE (see electronic supplementary material) [4,5]. The impact of the Yamnaya migration on the formation of the CWC complex continues to be debated ([4] with comments in [5,6]). Genetic signals of migration and admixture have been found in central Europe, Scandinavia, and to the east of the Baltic Sea [1,2,7–9]. However, the development took different paths in different regions and it is not clear exactly how the migrations affected the demographic development and population history at the fringes of the European continent [5,6].

In Sweden, the CWC complex has been labelled the (Swedish-Norwegian) Boat Axe or Battle Axe Culture (BAC). The material manifestation of BAC, starting around 3000/2800 BCE [10], was distributed in Scandinavia up to modern-day Middle Sweden and southern Norway and on the eastern side of the Baltic Sea up to the southwestern parts of Finland (e.g. [11–13]). How the BAC/CWC complex dispersed into the Baltic countries and Fennoscandia has been extensively debated (e.g. [12,14–18]), especially in association with a process of migration or of cultural diffusion and local development [11,12,16,19,20]. Earlier archaeological research viewed the BAC/CWC complex as a society with common cultural and social practices and stressed the uniformity of burial customs, pottery design and typology, and the boat-shaped battle axes (e.g. [12,15,21]). Recently, it has been argued that these views were too simplistic, and regional patterns and traits within the BAC/CWC complex have been highlighted (e.g. [4–6,11,22]).

While previous archaeogenetic research has studied individuals connected with the BAC/CWC cultural complexes in some regions around the Baltic Sea [1,2,7–9,23], the character of the migration patterns and the temporal and regional dynamics within the CWC area as well as the BAC introduction into Scandinavia are yet to be explored. To achieve a better understanding of these population dynamics and the BAC introduction to Scandinavia, we investigated the demographic development in the CWC area around the Baltic Sea, in the third millennium BCE, by sequencing DNA from 11 prehistoric individuals from (modern-day) Sweden, Estonia, and Poland. By comparing the genetic profiles of the newly sequenced individuals to individuals from different regions of the CWC area and from different kinds of cultural contexts, we investigate ancestry and admixture to paint a more detailed picture of the demographic processes that took place across the CWC area, with a specific focus on the onset and subsequent dynamics of the BAC in Scandinavia.

## 2. Results

### (a) Genome sequencing, quality assessment, and genetic results

We generated and analysed genome sequence shotgun data from 11 individuals originating in northeastern Europe dated to 3300–1660 cal BCE (electronic supplementary material, table S1). Five individuals were excavated from CWC contexts: two from Obłaczkowo, Poland, one from Karlova, Estonia, and two from the CWC-related BAC burial Bergsgraven in Linköping, Sweden. The six additional individuals were from other archaeological contexts in Sweden: five from megalithic burial structures primarily associated with Funnel Beaker

Culture (FBC) (two from Rössberga in Västergötland and three from Öllsjö in Scania) and one from a Pitted Ware Culture (PWC) context (Ajvide on Gotland). Radiocarbon dating showed that the three individuals from the Öllsjö megalithic tomb derived from later burials, where oll007 (2860–2500 cal BCE) overlaps with the time interval of the BAC, and oll009 and oll010 (1930–1650 cal BCE) fall within the Scandinavian Late Neolithic and Early Bronze Age (table 1; electronic supplementary material, table S1 and figure S2). Genome-wide sequence coverages range from 0.1 to 3.2×, and the sequence data for all individuals exhibit characteristic properties of ancient DNA: short fragment size and cytosine deamination at the ends of fragments (e.g. [24]) (table 1; electronic supplementary material, figure S5). Estimates of mitochondrial contamination [25] were low, less than 2% for all 11 individuals, as was the estimated nuclear contamination on the X-chromosome in males [26,27] (less than 1.2%) (table 1; electronic supplementary material, table S3). Five individuals were genetically determined to be males, and six were females, based on the fraction of sequence fragments mapping to the different sex chromosomes [28] (table 1).

The individuals from BAC and CWC contexts, including oll007 from a megalithic burial, displayed U4 and U5 mitochondrial DNA (mtDNA) lineages, previously associated with Stone Age hunter–gatherers [29–34], and H1, N1a, and U3 lineages, associated with Neolithic farmers [1,32,35,36] (table 1; electronic supplementary material, table S4). This broadly coincides with the wide variety of mtDNA lineages found in other individuals from CWC contexts (e.g. [2,32]). However, the U3 and N1a lineages, which were found here (poz44 and ber2), have not been reported from individuals excavated in CWC contexts. The two males in our dataset (ber1 and poz81) belonged to Y-chromosome R1a haplogroups (table 1; electronic supplementary material, table S5), as do the majority of males (16/24) from the previously published CWC contexts (Viby in Sweden, Ardu and Kunila in Estonia, Gyvakarai and Spiginas in Lithuania, Bergrheinfeld and Esperstedt in Germany, and Brandýsek in the Czech Republic) [1,2,7,31,32,37], while a smaller fraction belonged to R1b [3/24] or I2a [3/24] lineages (Tiefbrunn and Esperstedt in Germany, Pikutkowo and Łęki Małe in Poland, and Brandýsek in the Czech Republic) [2,23,32,37]. The R1a haplogroup has not been found among Neolithic farmer populations nor in hunter–gatherer groups in central and western Europe, but it has been reported from eastern European hunter–gatherers and Eneolithic groups [1,31,32]. Individuals from the Pontic–Caspian steppe, associated with the Yamnaya Culture, carry mostly R1b and not R1a haplotypes [1,2,31,32].

Three individuals had enough data for investigating the *LCT* gene-region (electronic supplementary material, table S6), and one of these individuals (kar1) carried at least one allele (-13910 C->T) associated with lactose tolerance, while the other two individuals (ber1 and poz81) carried at least one ancestral variant each, consistent with previous observations of low levels of lactose tolerance variants in the Neolithic [1,2,33,38] and a slight increase among individuals from CWC contexts [32]. The individuals further displayed a mixed appearance with both light and dark hair and brown and blue eyes (electronic supplementary material, table S6). Stable carbon and nitrogen isotope values for the individuals from modern-day Sweden show a terrestrial diet except for the Ajvide individual (electronic supplementary material, table S1 and figure S3). Strontium isotope data for the two

**Table 1.** Information on the 11 ancient individuals investigated in this study, including radiocarbon dates, average read length, average genome coverage, mtDNA coverage, mtDNA and Y-chromosome haplogroups, biological sex, and contamination estimates based on mtDNA and on the X-chromosome in males.

| sample | site/grave | country | context | date cal BCE (95% CI) | Av. RL | nu cov | MT cov | MT hg | Y hg | biol sex | cont est MT (%) | cont est X (%) |
|---|---|---|---|---|---|---|---|---|---|---|---|---|
| ber1 | Bergsgraven | Sweden | BAC | 2620–2470 | 84.5 | 3.24 | 1344 | U4c1a | R1a-Z283 | XY | 0.19 | 0.80 |
| ber2 | Bergsgraven | Sweden | BAC | 2640–2480 | 74.4 | 0.48 | 443 | N1a1a1 | — | XX | 0.26 | — |
| oll007 | Öllsjö | Sweden | Megalithic[a] | 2860–2500 | 77.4 | 1.24 | 86 | H1c | — | XX | 0.44 | — |
| oll009 | Öllsjö | Sweden | Megalithic[a] | 1930–1750 | 99.7 | 1.01 | 325 | H6a1b3 | n.a. | XY | 1.97 | 0.32 |
| oll010 | Öllsjö | Sweden | Megalithic[a] | 1880–1660 | 87.7 | 0.26 | 96 | X2b11 | — | XX | 1.83 | — |
| kar1 | Karlova | Estonia | CWC | 2440–2140 | 61.9 | 2.35 | 2481 | H1f1a | — | XX | 0.79 | — |
| ajv54 | Ajvide | Sweden | PWC | 2900–2680 | 89.3 | 0.91 | 510 | U5b1d2 | n.a. | XY | 1.24 | 0.58 |
| ros3 | Rössberga | Sweden | FBC | 3330–2930 | 64.1 | 0.37 | 30 | K1b1a1 | — | XX | 0.40 | — |
| ros5 | Rössberga | Sweden | FBC | 3090–2920 | 97.9 | 0.85 | 106 | J1c5 | I-M429* | XY | 0.19 | 0.28 |
| poz44 | Obłaczkowo | Poland | CWC | 2870–2580 | 86.6 | 0.11 | 253 | U3a'c | — | XX | 0.29 | — |
| poz81 | Obłaczkowo | Poland | CWC | 2880–2630 | 84.4 | 1.87 | 172 | U4b1b2 | R1a-M417 | XY | 1.32 | 1.15 |

[a]Megalithic contexts mean that the individuals are from the FBC-associated megalithic tomb, but constitute secondary burials and have radiocarbon dates associated with the BAC time period (oll007) or the Scandinavian Late Neolithic and Bronze Age (oll009 and oll010).

individuals in Bergsgraven have differing signals, indicating recent migration of at least one of the individuals to the area (electronic supplementary material, table S2 and figure S4).

## (b) Exploratory analyses

To investigate the genomic ancestry of the 11 individuals from this study in context with other ancient individuals, we projected them onto the PC1–PC2 space of 60 modern-day Western Eurasian populations (figure 1a) [40] together with 272 relevant prehistoric individuals who have been previously reported (electronic supplementary material, table S7). We further used an unsupervised clustering approach [41] to examine the broad-scale genomic affinities (electronic supplementary material, figure S6).

The broad continental pattern that appears confirms previous results (figure 1a) [3]: (i) a clear separation between Early and Middle Neolithic farmers and Mesolithic hunter–gatherers from all over Europe [1,30,32–34,42,43], (ii) substructure among the hunter–gatherers roughly corresponding to an east–west gradient [1,8,29,31,32], with an exception in Scandinavia [29], and (iii) most Late Neolithic and Bronze Age individuals overlapping with modern-day central and northern Europeans due to admixture with incoming groups related to the Yamnaya herders from the Pontic steppe [1,32].

In Scandinavia, we see a clear separation between individuals from the three different archaeological complexes, the FBC, PWC, and BAC, with individuals associated with the farming FBC (including the Rössberga individuals reported here) grouping with other Middle Neolithic farmer groups (figure 1a). Neolithic foragers associated with the Pitted Ware Culture (including ajv54) were genetically similar to Mesolithic Scandinavian hunter–gatherers with some trend towards farming groups, likely due to admixture between the two groups [9,29,33]. The individuals associated with BAC (including the two individuals from Bergsgraven) show a clear association with individuals from other CWC contexts elsewhere in Europe. Notably, the oll007 individual from the megalithic site at Öllsjö (southern Sweden), who was not directly associated with any CWC-related artefacts but overlaps with CWC chronologically, clusters with individuals from CWC contexts, as do the two individuals with later dates (oll009 and oll010) from the same site.

Similar to the BAC individuals, the newly sequenced individuals from the present-day Karlova in Estonia and Obłaczkowo in Poland appear to have strong genetic affinities to other individuals from BAC and CWC contexts across the Baltic Sea region [1,2,7,8,23,31,32] (figure 1a). Some individuals from CWC contexts, including the two from Obłaczkowo, cluster closely with the potential source population of steppe-related ancestry, the Yamnaya herders. Notably, these individuals appear to be those with the earliest radiocarbon dates among all genetically investigated individuals from CWC contexts. Overall, for CWC-associated individuals, there is a clear trend of decreasing affinity to Yamnaya herders with time (figure 2).

## (c) Admixture modelling of Corded Ware Culture

To investigate the spread of CWC-related people into Scandinavia and obtain a clearer picture of the ancestry proportions found in individuals associated with the CWC and the BAC, we conducted a supervised modelling of their ancestry from three sources: Anatolian farmers, western European hunter–

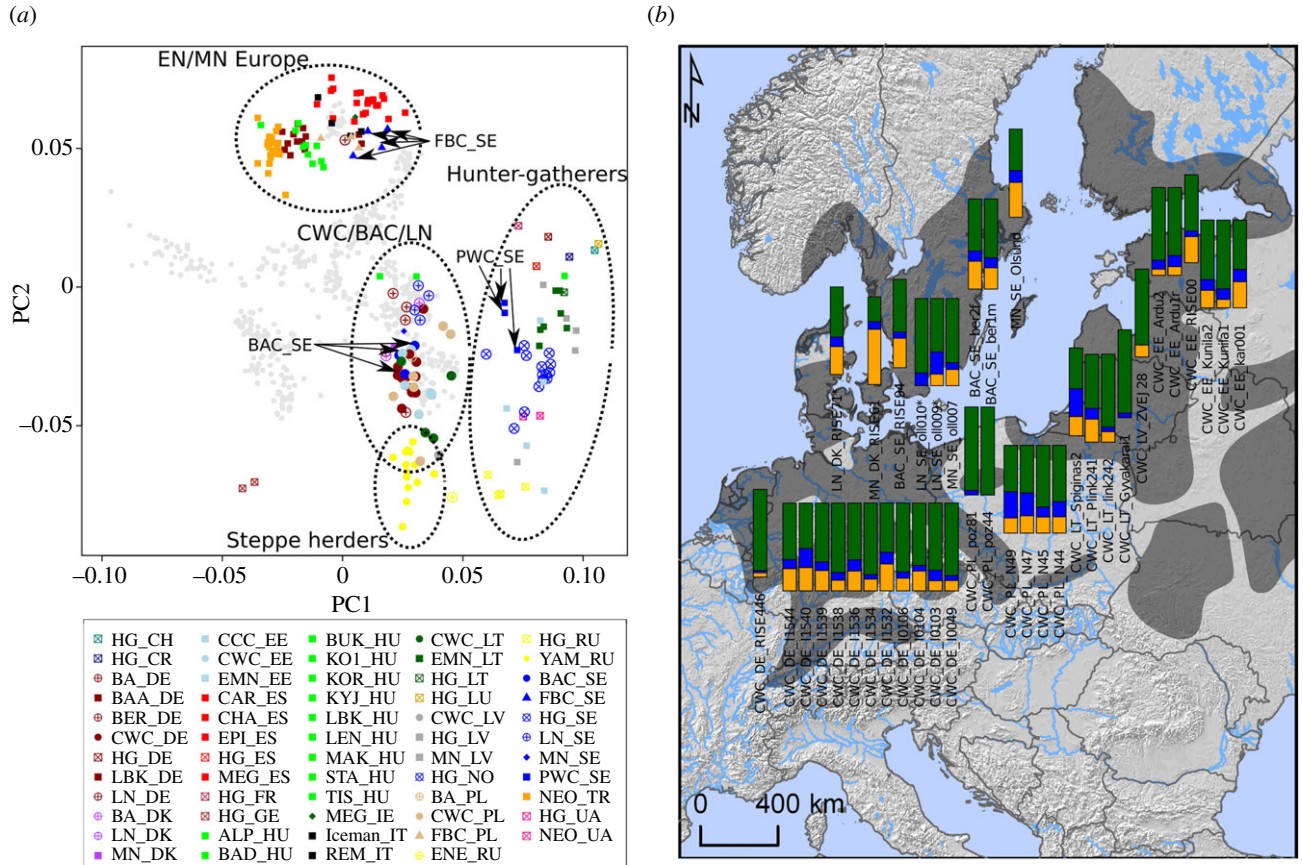

**Figure 1.** (*a*) Principal component analysis of modern Europeans (grey) and projected ancient Europeans. The full list of ancient samples with sample IDs, the associated culture or time period followed, and two-letter International Organization for Standardization codes for country can be found in table 1 and electronic supplementary material, table S7. (*b*) Geographical range of BAC/CWC and admixture modelling with qpAdm of BAC/CWC individuals in a three-source model: Anatolian Neolithic (orange), European hunter–gatherers (blue), and Yamnaya steppe herders (green). Individuals marked with an asterisk lived after the BAC/CWC time period. Map source: Sjögren *et al.* [39], published CC-BY.

gatherers, and Yamnaya steppe herders. We estimated ancestry under this three-source model per individual using qpAdm [1]. The geographical pattern of ancestry is shown in figure 1b. This approach confirms that most ancestry in all CWC individuals originates from a group related to Yamnaya steppe herders (green), while smaller contributions from Western hunter–gatherers (blue) and Neolithic farmers (orange) are also widespread (figure 1b). Individuals from BAC contexts in (modern-day) Sweden also show this pattern, with a large genetic component tracing its ancestry to Yamnaya steppe herders and with small ancestry components related to Neolithic farmers and hunter–gatherers. The other individuals who were contemporary with BAC but had unclear cultural contexts, and who were buried in the Ölljsö megalith constructed many hundred years earlier (oll007), or found as a stray find (Ölsund) [9], show the same genetic profile as individuals from typical BAC contexts in other parts of Sweden. Although the individuals associated with the BAC (including the oll007, oll009, oll010, and Olsund) have a large proportion of steppe ancestry, it is relatively low compared with most other individuals from CWC contexts. The female (kar1) from the Karlova CWC burial in Estonia shows similar patterns, also displaying slightly less steppe ancestry than other CWC-associated individuals from Estonian sites [2,7]. By contrast, the CWC individuals from Obłaczkowo in Poland (poz44 and poz81) show an extremely high proportion of steppe ancestry (greater than 90%), which is different from the later CWC-associated individuals

excavated in Pikutkowo (Poland) [23], but similar to some other CWC-associated individuals from Germany, Lithuania, and Latvia [2,8,31]. Interestingly, these individuals with a large fraction of steppe ancestry have typically been dated to more than 2600 BCE, making them among the earliest CWC individuals genetically investigated. This observation, i.e. early CWC individuals resembled (genetically) Yamnaya-associated individuals, while later CWC groups show higher levels of European Neolithic farmer ancestry (Pearson's correlation coefficient: −0.51, *p* = 0.006) (figure 2), suggests an initial dispersal that occurred rapidly.

## (d) Admixture modelling of Battle Axe Culture

The positioning in the principal component analysis (PCA) (figure 1a) and the admixture modelling results (figure 1b) suggest ancestry from three different main ancestral groups in BAC: Anatolian farmers, European hunter–gatherers, and Yamnaya steppe herders. Direct candidates for contributions to BAC are the Scandinavian Middle Neolithic PWC and FBC plus a third population carrying high proportions of Yamnaya-related ancestry (YAM), like CWC. We used an explicit model to investigate contributions from predefined ancestral groups (qpWave of ADMIXTOOLS) [1,44] to model the BAC genetic make-up. We found that the BAC-associated individuals can be modelled as a combination of genetic material from these three groups (*p* = 0.80). In addition to the three-source model, two models only

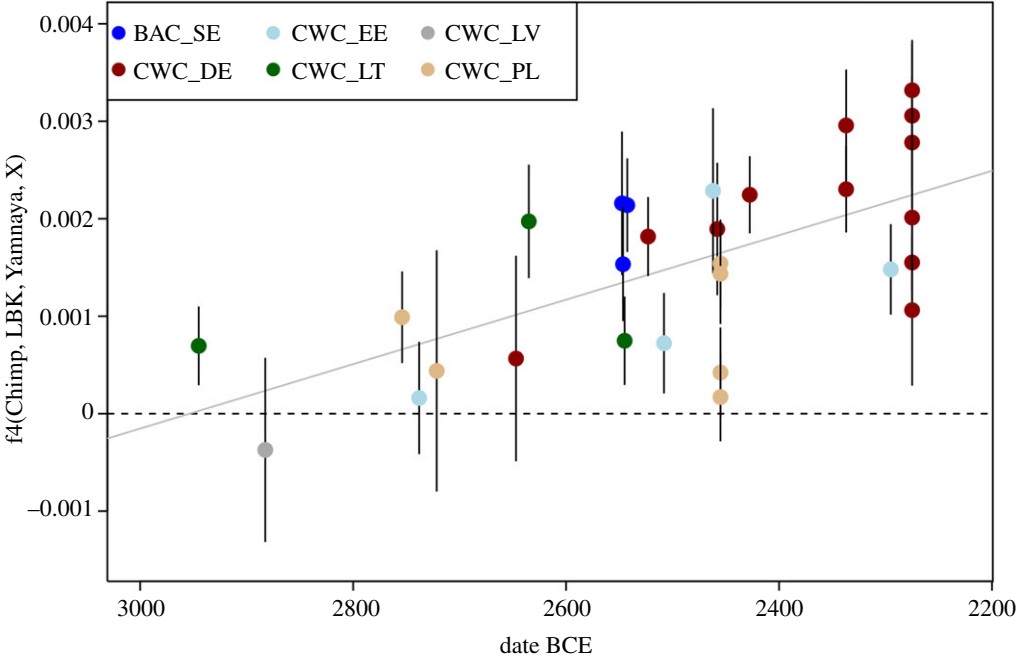

**Figure 2.** Correlation between f4(Chimp, LBK, YAM, X), where X is a CWC or BAC individual, and the date (BCE) of each individual (table 1 and [1,2,7,8,23,31,32,37]). This statistic measures shared drift between CWC and Linear Pottery Culture (LBK) as opposed to YAM and should increase with the higher proportion of Neolithic farmer ancestry in CWC and BAC. The pattern is not driven by spurious affinities between single nucleotide polymorphism (SNP) capture or shotgun data (electronic supplementary material, figures S9 and S10). Ages on the x-axis correspond to the mid-point of the interval for the date of each sample as reported in their original publication (electronic supplementary material, tables S7 and S8). (Online version in colour.)

using two sources were consistent with the data: the best fitting two-source model was FBC + YAM ($p = 0.86$), while YAM + PWC would also fit the data (although trending toward low $p$-values; $p = 0.07$). These observations suggest that—to our statistical resolution—a direct PWC contribution to BAC is not needed in a model, but actual PWC admixture might have been small or there may have been indirect PWC contributions through PWC first mixing with FBC [34] who later contributed ancestry to BAC. Notably, using only the CWC population as the single source for BAC was consistent with the data in all cases ($p > 0.05$, except when using CWC-associated individuals from Latvia, CWC_LV), showing that the BAC individuals in Sweden could not have emerged without migrations from other CWC groups.

To find out if BAC represents an admixed group that came into Scandinavia or if (at least some of) the admixture took place in Scandinavia, we investigated the relationship among the Scandinavian BAC group and various CWC groups from other geographical areas by constructing admixture graphs (figure 3). The BAC groups fit as a sister group to the CWC-associated group from Estonia (CWC_EE, electronic supplementary material, figure S8) but not as a sister group to the CWC groups from Poland (CWC_PL, figure 3) or Lithuania (CWC_LT, electronic supplementary material) ($|Z| > 3$), indicating some differences in ancestry between these CWC groups and BAC (CWC from Latvia, CWC_LV, and the CWC group from Germany, CWC_DE, had to be excluded from this analysis due to the number of SNPs being too low, see electronic supplementary material). Supervised admixture modelling suggests that BAC may be the CWC-related group with the lowest YAM-related ancestry and with more ancestry from European Neolithic groups (figure 1b). Consequently, models, where BAC is an admixed group between a CWC group and additional ancestry from a group carrying ancestry-related European Neolithic groups,

were consistent with the data for all three tested CWC groups ($|Z| < 2.3$, figure 3; electronic supplementary material, figures S7 and S8).

Two potential scenarios were consistent with the data to explain the emergence of BAC in Scandinavia: (i) direct migration of CWC groups from the eastern Baltic or (ii) a migration of CWC groups from the southern Baltic Sea region mixing with FBC groups in Scandinavia. Future archaeogenetic studies could fill geographical and chronological gaps by including more samples that should help to distinguish between the scenarios.

## 3. Discussion

The gene pool of modern Europeans has been shaped by a number of prehistoric events and migrations. The CWC horizon represents one of the major demographic processes in the northern areas of Europe as this is associated with the first occurrence of steppe-related ancestry. Our results have implications for our view on the demographic development associated with the CWC in general, and the Scandinavian variety of the BAC specifically.

People buried in CWC contexts display a genetic ancestry component that was not present in northern and central Europe prior to the third millennium BCE. This ancestry component, often called 'steppe ancestry', probably traces back to the 'Yamnaya expansion' of herders associated with the Yamnaya Culture that dispersed into Eastern Europe from the Pontic–Caspian steppe around 3000 BCE [1,2,32]. This component makes up the largest proportion of the genetic ancestry in all sequenced BAC/CWC individuals around the Baltic Sea: from the modern-day countries Estonia, Latvia, and Lithuania in the east; Poland and Germany in the southwest; and Denmark and Sweden in the northwest (figure 1b). One important observation is that the earliest

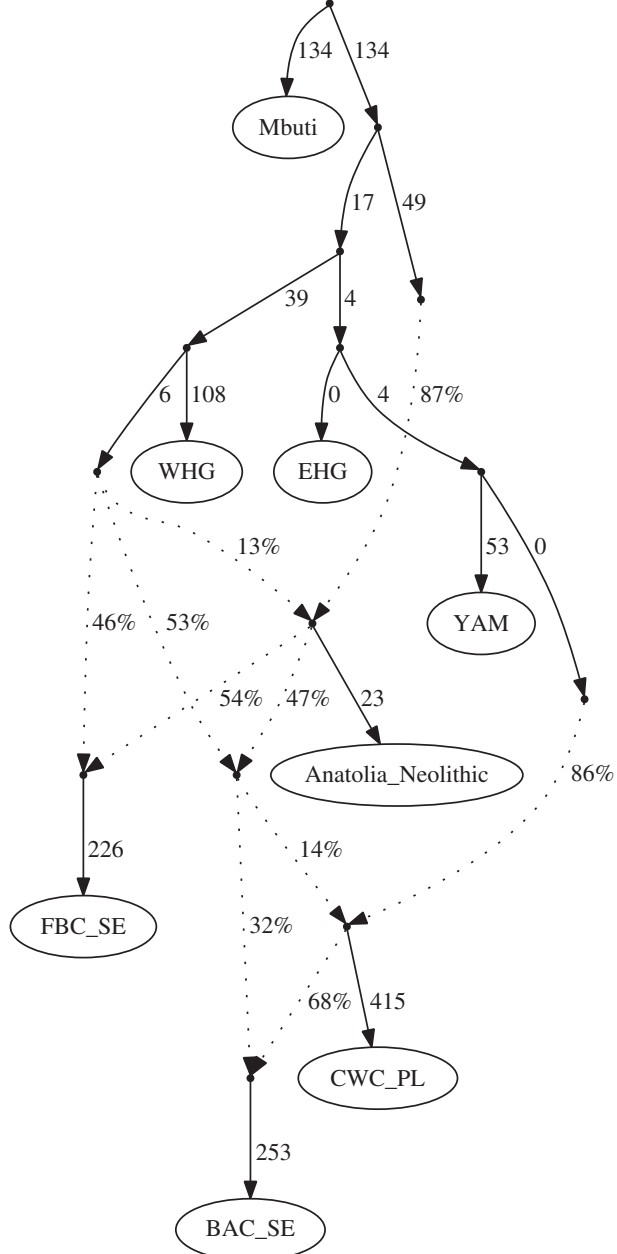

**Figure 3.** Admixture graph modelling BAC as a mix of a CWC source and an admixed Middle Neolithic group (worst |Z| < 1.68). CWC individuals from modern-day Poland (CWC_PL) chosen here, other groups shown in electronic supplementary material, figures S7 and S8. WHG, western hunter-gatherers; EHG, eastern hunter-gatherers. The sub-Saharan African Mbuti were used as an outgroup for this analysis.

CWC individuals analysed to date had the highest proportion of steppe ancestry (greater than 90%), while this proportion decreased in later individuals (figure 2). This suggests a gradual process of admixture between incoming groups and local groups, such as the FBC groups in northern Europe that traced most of their genetic ancestry to Anatolian Neolithic farmers. This process was driven by incoming males mixing mainly with local females [5,7,45,46]. The admixture process is evident across the entire distribution of the CWC, even in regions such as the eastern Baltic coast where no FBC groups or genetically related groups have been found. Consequently, farmer-related ancestry must have arrived in the eastern Baltic region via migrations more recent than 2500 BCE [7–9]. Possibilities include a generally large exchange network across the entire CWC horizon or specific migrations into the eastern Baltic region. Potential

source regions for the latter could be modern-day Poland or Sweden, where FBC groups predating the arrival of the CWC are found.

The paternal lineages found in the BAC/CWC individuals remain enigmatic. The majority of individuals from CWC contexts that have been genetically investigated this far for the Y-chromosome belong to Y-haplogroup R1a, while the majority of sequenced individuals of the presumed source population of Yamnaya steppe herders belong to R1b [1,2]. R1a has been found in Mesolithic and Neolithic Ukraine [1,31,32]. This opens the possibility that the Yamnaya and CWC complexes may have been structured in terms of paternal lineages—possibly due to patrilineal inheritance systems in the societies [47,48]—and that genetic studies have not yet targeted the direct sources of the expansions into central and northern Europe.

The Scandinavian Middle Neolithic megalithic tombs are associated with the FBC. However, their reuse, indicated by artefacts common to the BAC and later periods, has been noted [17]. The oll007 individual, buried in the FBC-associated Öllsjö megalithic tomb, but radiocarbon dated to the time period of the BAC, is genetically very similar to individuals from BAC contexts (e.g. Bergsgraven and Viby). Thus, although archaeologically the reuse of megalithic tombs was assumed earlier [17], our study may be the first direct link (using genetics) showing that indeed FBC-associated megalithic tombs were used as burial places also for the people of the BAC. This could possibly also extend to the Danish Single Grave Culture (SGC) [49], as RISE61 [2], a male buried in the Kyndeløse passage grave and with a radiocarbon date overlapping with the BAC/CWC/SGC time period, also displays some steppe ancestry.

The BAC replaces the FBC in the southern parts of Scandinavia and was previously assumed to have been a cultural adaptation of existing groups [12,17]. We show in multiple individuals from different parts of Scandinavia that these groups (BAC) were part of the general CWC horizon, i.e. they too are the result of admixture of different groups tracing parts of their ancestry to European hunter–gatherers, Anatolian Neolithic farmers, and Yamnaya steppe herders. This implies that BAC groups were not the direct descendants of any of the groups that lived in the area previously or even contemporaneously—i.e. the groups associated with FBC or the PWC. We also note that the BAC group does not have a particular genetic connection with other eastern Baltic groups such as the Combed Ceramic Culture. The mixed ancestry of individuals in BAC contexts is evident across all autosomal analyses, as well as in mitochondrial haplogroups, but the paternal haplogroups stand out to some extent, showing a deviant, more extreme, pattern. This Y-chromosome pattern is, however, consistent with a male sex-biased migration and admixture process among the Yamnaya, and later CWC, groups [7,39,45,50].

As the individuals of the BAC complex cannot be modelled as direct genetic descendants of FBC or PWC groups, a migration into Scandinavia of people with a large proportion of steppe ancestry must have taken place. We were not able to find unambiguous evidence for a specific source population by testing all other individuals associated with the CWC that have been genomically investigated as potential ancestors. The Scandinavian BAC group has more Neolithic farmer ancestry than pre-2600 BCE individuals in CWC contexts from the southern or eastern Baltic coast, suggesting that they mixed with an FBC group. This process

could have happened in Scandinavia or before arriving in Scandinavia. On the one hand, the CWC people were mainly bound to travels on land, which would favour migration from central Europe into modern-day Denmark and Sweden (e.g. [16]). On the other hand, there is evidence for technological exchange (pottery craft) crossing the Baltic Sea in this time period [20,51], which may or may not be associated with gene flow. Finally, the genetic data from BAC contexts are still limited, and the patterns of gene flow that we observe are consistent with both a single migration event into Scandinavia and with a continuous process with an extensive network of social and technological exchange. Future studies will refine our understanding of the social, geographical, and temporal dynamics during this important period in European prehistory.

## 4. Material and methods

We extracted DNA, prepared Illumina DNA libraries [29,33,34], and shotgun sequenced DNA from 11 archaeological remains from Sweden, Estonia, and Poland (table 1; electronic supplementary material). The samples were radiocarbon dated ($n = 10$) and analysed for stable isotope values (carbon, $n = 8$; nitrogen, $n = 8$; strontium, $n = 2$). Sequence data were merged, trimmed, and mapped to the human genome, and the data were authenticated using fragmentation, deamination patterns, and various contamination estimates [29]. Mitochondrial and Y-chromosomal DNA haplogroups were called for the newly sequenced individuals. The comparative data consisted of ancient individuals with at least 0.1× genome coverage who lived at the time of the CWC or before (electronic supplementary material, table S7) [1,2,7–9,23,29–34,42,43,52–59] and modern populations from the Human Origins dataset [30] and the public data of the Simons Genome Diversity Panel [60]. Ancestry and admixture were assessed using principal component analyses, unsupervised ADMIXTURE [41], and admixture

proportions were estimated using ADMIXTOOLS' qpWave and qpAdm [1,44], and phenotypic variation was investigated [61,62]. More detailed descriptions of the archaeological samples and sites and the methods used can be found in the electronic supplementary material.

Data accessibility. The genome data for the 11 investigated individuals are available at the European Nucleotide Archive (accession number PRJEB32786).

Authors' contributions. H.M., T.G., A.G., J.S., and M.J. designed the research; H.M., T.G., E.M.S., A.J., M.F., A.R.M., A.G., J.S., and M.J. performed the research; H.M., A.J., M.F., Ł.P., M.T., J.L., and J.S. contributed samples and conducted archaeological analyses; H.M., T.G., and E.M.S. analysed data; and H.M., T.G., A.G., J.S., and M.J. wrote the paper with input from all authors.

Competing interests. We declare we have no competing interests.

Funding. This work was supported by the Berit Wallenberg Foundation (grant no. BWS2011.0090 (M.F.)); the Swedish Research Council (grant nos 2017-02503 (H.M.), 2017-05267 (T.G.), 2013-1905 (M.J., J.S., and A.G.)); Riksbankens Jubileumsfond (grant no. M13-0904:1 (M.J., J.S., and A.G.)) and Knut and Alice Wallenberg foundation (M.J., J.S., and A.G.). The Ministry of Science and Higher Education of the Republic of Poland funded the radiocarbon dates for poz44 and poz81 (grant no. N N109 287 137 (L.P.)).

Acknowledgements. We thank the following museums and persons for supplying bone material: P. Nilsson, M. Ohlsén, P. Nyberg, and M. Douglas from Östergötlands Museum, L. Drenzel and J. Karlsson at the National Historical Museums (Stockholm), the Historical Museum at Lund University (LUHM), P. Pawlak from Henryk Klunder's Pracownia Archeologiczno-Konserwatorska (Poznań) and Ü. Tamla and the Archaeological Research Collections, Tallinn University, Estonia. Further, we thank J. Evans at NERC, Isotope Geosciences Laboratory, Nottingham, UK, for performing strontium isotope analyses and T. Hedlund for osteological analyses on the Öllsjö individuals. Shotgun sequencing was performed at the National Genomics Infrastructure (NGI) Uppsala, and computations were performed at Uppsala Multidisciplinary Center for Advanced Computational Science (UPPMAX) under projects b2013203, b2013240, b2015066, snic2017-7-259, sllstore2017020, and sllstore2017087.

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
