## [Reviewer comments · Proceedings of the Royal Society B: Biological Sciences]

Review History

RSPB-2019-1528.R0 (Original submission)

Review form: Reviewer 1

Recommendation

Accept with minor revision (please list in comments)

Scientific importance: Is the manuscript an original and important contribution to its field?

Excellent

General interest: Is the paper of sufficient general interest?

Good

Quality of the paper: Is the overall quality of the paper suitable?

Good

Is the length of the paper justified?

Yes

Should the paper be seen by a specialist statistical reviewer?

No

Do you have any concerns about statistical analyses in this paper? If so, please specify them explicitly in your report.

No

It is a condition of publication that authors make their supporting data, code and materials available - either as supplementary material or hosted in an external repository. Please rate, if applicable, the supporting data on the following criteria.

Is it accessible?

No

Is it clear?

Yes

Is it adequate?

Yes

Do you have any ethical concerns with this paper?

No

Comments to the Author

This manuscript reports new ancient DNA evidence, in form of whole genome sequence data, from eleven individuals from North and Northeast Europe whose skeletal remains were buried in the context of distinct Middle and Late Neolithic cultures. The main focus of the paper is on Scandinavian Battle Axe Culture (BAC) which culturally belongs to the sphere of the wide-spread Corded Ware Culture (CWC). Previous aDNA studies have shown that burials within the CWC context draw majority of their genetic ancestry from the Steppe Belt region. It has not been clear, however, to what extent BAC represents the spread of CWC to Scandinavia 'as in association with a process of migration or of cultural diffusion and local development'. The genomes of two Swedish BAC individuals analysed in context of the genomes of the earlier Funnel Beaker Culture (FBC) and other Neolithic genomes analysed by the authors and those reported in the literature showed that BAC spread to Scandinavia together with a strong genetic component and with an indication of some level of admixture with FBC, with time series analyses revealing more admixture occurring later on in the Late Neolithic, which is consistent with the trends in Europe reported earlier in the literature.

Given the scarcity of genomic data from Neolithic North and Northeast Europe and questions surrounding population origins and continuity I find that this is an important contribution to the field. Of particular interest is the finding of a Megalithic burial with genetic affinity to BAC genomes but without any cultural associations. The methods to detect and quantify genetic ancestry proportions in the data seem adequate for the purpose and I do not see any major problems with the analyses. However, there are some minor issues I wanted to highlight, particularly with regards to the clarity of the statements and the questions set.

1. Abstract 'The BAC groups likely mixed with resident middle-Neolithic farmers (e.g. FBC groups), but not with Neolithic foragers, revealing social stratifications in the Scandinavian Neolithic.' This statement is not exactly in line with the (2d) admixture modelling of BAC section where the YAM+PWC forager model is although less supported than the other but not rejected. So, we cannot rule out such contributions on the basis of the results reported here. Furthermore, I am not sure what exactly could be deduced about the social stratifications. No analyses or statements on this theme are presented in the results section and the discussion ends with a suggestion that 'Future studies will refine our understanding of the social, geographic, and

temporal dynamics during this important period in European prehistory.' So, it seems the statement in the abstract about social stratifications in the Scandinavian Neolithic is premature and not supported either. I suggest the authors to remove this sentence from the abstract.

2. While the research objectives and questions are implicitly made clear in the introduction there is a certain degree of vagueness throughout the manuscript about these. E.g. (but not limited to these few examples): 'We present an archaeogenetic investigation' - in the abstract, and, 'we investigate the demographic development in the CWC area', 'we paint a detailed picture' in the intro, instead of clearly set up question(s); I suggest the authors to revise the manuscript accordingly. Similarly, the more specific objectives of particular analyses should be presented more clearly: e.g. (2b) 'To place the 11 individuals from this study into context' or (2c) 'To obtain a clearer picture of the ancestry' - these were probably not really the end goals.

3. Figure 2. Firstly, given that these individual genomes come from a wide geographic area, the authors may consider distinguishing the geography (either by symbols or by colour) on this plot. Secondly, given that the f4 statistic that is shown, $f_4(\text{Chimp}, \text{LBK}; \text{YAM}, \text{X})$, does not explicitly include any hunter-gatherer genomes, '.. higher proportion of hunter-gatherer and Neolithic farmer ancestry in CWC and BAC.' is not clear as how the higher hunter-gatherer contributions should be read out from the graph. I suggest either to clarify this or remove the word 'hunter-gatherer' from this context.

Review form: Reviewer 2

Recommendation

Accept with minor revision (please list in comments)

Scientific importance: Is the manuscript an original and important contribution to its field?

Good

General interest: Is the paper of sufficient general interest?

Good

Quality of the paper: Is the overall quality of the paper suitable?

Excellent

Is the length of the paper justified?

Yes

Should the paper be seen by a specialist statistical reviewer?

No

Do you have any concerns about statistical analyses in this paper? If so, please specify them explicitly in your report.

No

It is a condition of publication that authors make their supporting data, code and materials available - either as supplementary material or hosted in an external repository. Please rate, if applicable, the supporting data on the following criteria.

Is it accessible?

No

Is it clear?

N/A

Is it adequate?

N/A

Do you have any ethical concerns with this paper?

No

Comments to the Author

Malmström et al. 2019 present a simple but insightful archaeogenetic investigation of the demographic origins of individuals associated to the Battle Axe Culture (BAC). By analyzing the genomes the authors reports in their study, together with previously published data, Malmström et al. 2019 show that the BAC represents a group different from earlier populations in Scandinavia. Previous archaeological hypothesis suggested that the BAC complex was a continuation and transformation of the Funnel Beaker Culture (FBC), prehistoric farmers who inhabited southern Sweden.

In addition, this manuscript further builds on demographic dynamics of the arrival of the “steppe-like” ancestry to Neolithic Europe, via Corded Ware Culture (CWC) expansion. Authors find that there is a significant decrease “steppe-like” ancestry with time in CWC/BAC individuals, suggesting that while the initial migration occurred rapidly, later CWC/BAC individuals admixed with local inhabitants of the area.

Overall, the manuscript presented by authors is very well written, thorough in its methodology and sound with its conclusions. Therefore I recommend the publication of this article. However, in order to improve the quality and readability of this manuscript, I suggest the following issues are addressed:

1) Change the “Adimxture Graph” Electronic Supplementary Material (ESM) subsection to “f4 statistic and Adimxture Graph” p.e.

It took me some time to find the one line of text, describing how the f4 analysis was done since it was in a section with a different subheading.

2) In the “Sample preparation” ESM section it mention that ...“some libraries were enriched using Caucasian baits and Mybait Human Whole Genome Capture Kit (MYcroarray) following the manufacturer’s instructions (Mybaits manual version 2.3.1).”

2.1 I find the term “Caucasian baits” unscientific, uninformative, and unfortunate that company refers to it with such a name. Assuming that the name cannot be modified because of a legal brand issue, could authors at least quote the term as above to denote the controversy of using that term (i.e. “Caucasian baits”)?

2.2 As stated above “some libraries” were enriched using capture enriching techniques.

However, it is not specified anywhere in the text, to which individuals does this libraries belong to, how much data was produced from those libraries vs blunt-end ones, nor if they were included in the demographic analyses.

3) Individual’s displayed on the x-axis of Fig. 2 represent the point estimates of samples 95% C.I. calibrated dating, an average of the C.I., the upper bound of the C.I. or the lower bound of the C.I.?

In the text where Fig. 2 is being referred, it says “Overall, for CWC associated individuals, there is a clear trend of decreasing affinity to Yamnaya herders with decreasing absolute age (figure 2).” What does “absolute age mean”? Please specify this term, as well as correct the meaning of the dating on the x-axis of Fig. 2, either from Fig. 2 legend or the ESM.

4) The manuscript and ESM makes reference to dates overall with a “BCE” nomenclature, however the archeological description of the Estonian “Karlova” site in the ESM text, as well as Table S1 uses the “BP” nomenclature. Please use the same archaeological dating style consistently. I suggest sticking to the former, and make sure that all instances where an archaeological date is displayed is not using non-updated “BP” dates.

5) Add the geographical/cultural labels from Fig. 1B to Fig. 2

In general, I like the design and display of figures and tables from the paper. However I think that the reader will benefit if authors provided the the geographical/cultural labels from Fig. 1B to the f4 “decrease of steppe-like ancestry in time” Figure 2, to display the genetic affinities, age and place of origin/culture of investigated individuals in the same figure.

The supplementary table where the ancient DNA (aDNA) reference data is reported (Table S7), doesn't have the individual's dates. While the reader could infer the date of individuals analyzed in Fig. 2, by contrasting the information from Fig. 1B and Fig.2 this is not straight forward.

Thus, in order to more easily convey the results of the paper, as well as to allow the reader to further explore the temporal/geographical distribution of steppe-like ancestry in CWC, I suggest this change is implemented.

6) Yamnaya genetic affinities potentially affected by “data-generated methodological bias” in Fig. 2?

Malmström et al. 2019 test the f4(Chimp, LBK, Yamnaya, X) topology to measure the shared drift between BAC/CWC and Linear Pottery Culture (LBK), as opposed to YAM. Shared genetic drift should increase with higher proportion of hunter-gatherer and Neolithic farmer ancestry in CWC/BAC.

However, this test is potentially affected by using shotgun and captured generated-data in a direct comparisons. Non-captured and captured generated data have different read mapping biases, and directly comparing the shared drift of aDNA data generated with both different strategies can display affinities due to a shared experimental methodology and not to a recent shared demographic history.

According to Table S7 it seems “LBK” individuals are mostly captured-generated (except for the “Stuttgart” sample), while the Yamnaya individuals were both captured and non-captured generated (RISE and I0xx individuals). On one hand, if YAM data was composed mostly of captured-generated data, then the test could be biased against non-captured generated by an artificial affinity signal between the LBK and YAM, which is a conservative scenario.

For instance, there is a “X” sample in Fig. 2 dated around ~ 2900 BCE for which the point estimate and lower SE bound suggests that LBK and YAM are closer to each other, than LBK to “X”. In order words, “X” would be an outgroup to both. This is an unexpected demographic scenario, since if “X” was 100% YAM one should see that LBK is equidistant to both YAM and X. As authors explain, shared drift of “X” and LBK should increase with higher proportion of hunter-gatherer and Neolithic farmer ancestry in CWC/BAC. While the results of this test is not significantly negative, a shift towards negative values suggest an artificial signal is partly influencing this result.

On the other hand, if if YAM data was composed mostly of non-captured data and “X” was captured-generated data one potentially observed a “exaggerated” affinity between LBK and “X” just because that data was generated in the same way, although a real demographic signal might exist. In the worst case scenario this could create an artificial decrease of YAM ancestry signal with time in CWC/BAC individuals.

Having said that, “methodological bias” has a higher influence when the data being directly compared is from similar demographic background, which LBK and YAM are not.

Nonetheless, in order to improve the readability of the paper and so that future aDNA projects can further understand the effect of “methodological bias” in direct genetic comparisons, I suggest that authors add the geographical/cultural labels recommended in 5), so that the reader can assess whether captured and non-captured generated data show differences in YAM genetic affinity given the used topology.

Furthermore, I suggest that authors include the discussion above into the suggested “f4 statistic” section (see #1) of the ESM, so that the reader has more information on the matter. Such section will benefit also from the authors reporting how much data is captured and non-captured from both the LBK and YAM individuals with a small table or described in the text.

Finally, if needed and if there is enough data, authors could repeat the f4 analysis in Fig. 2 using only captured-generated or non-captured generated at the time for both the LBK, YAM groups and “X” tested sample.

Decision letter (RSPB-2019-1528.R0)

14-Aug-2019

Dear Dr Guenther:

Your manuscript has now been peer reviewed and the reviews have been assessed by an Associate Editor. The reviewer's comments (not including confidential comments to the Editor) and the comments from the Associate Editor are included at the end of this email for your reference. As you will see, the reviewers and the Editors have raised some concerns with your manuscript and we would like to invite you to revise your manuscript to address them.

When submitting your revision please upload a file under "Response to Referees" - in the "File Upload" section. This should document, point by point, how you have responded to the reviewer's and Editor's comments, and the adjustments you have made to the manuscript. We require a copy of the manuscript with revisions made since the previous version marked as "tracked changes" to be included in the "response to referees" document.

Research ethics:

Use of animals and field studies:

Please submit a copy of your revised paper within three weeks. If we do not hear from you within this time your manuscript will be rejected. If you are unable to meet this deadline please let us know as soon as possible, as we may be able to grant a short extension.

Best wishes,
Victoria Braithwaite

Professor V A Braithwaite
mailto: proceedingsb@royalsociety.org

Associate Editor, Comments to Author:

Two reviewers have submitted thoughtful, thorough reviews and both are positive about the subject and about the work. However, both reviewers make a number of suggestions for small but important changes that will make the manuscript clearer and, importantly, more accurate. The authors should pay careful attention to these and justify carefully where any of them have not / cannot be addressed.

=====

Reviewer's Comments to Author:

Referee: 1

This manuscript reports new ancient DNA evidence, in form of whole genome sequence data, from eleven individuals from North and Northeast Europe whose skeletal remains were buried in the context of distinct Middle and Late Neolithic cultures. The main focus of the paper is on Scandinavian Battle Axe Culture (BAC) which culturally belongs to the sphere of the wide-spread Corded Ware Culture (CWC). Previous aDNA studies have shown that burials within the CWC context draw majority of their genetic ancestry from the Steppe Belt region. It has not been clear, however, to what extent BAC represents the spread of CWC to Scandinavia ?as in association with a process of migration or of cultural diffusion and local development?. The genomes of two Swedish BAC individuals analysed in context of the genomes of the earlier Funnel Beaker Culture (FBC) and other Neolithic genomes analysed by the authors and those reported in the literature showed that BAC spread to Scandinavia together with a strong genetic component and with an indication of some level of admixture with FBC, with time series analyses revealing more admixture occurring later on in the Late Neolithic, which is consistent with the trends in Europe reported earlier in the literature.

Given the scarcity of genomic data from Neolithic North and Northeast Europe and questions surrounding population origins and continuity I find that this is an important contribution to the field. Of particular interest is the finding of a Megalithic burial with genetic affinity to BAC genomes but without any cultural associations. The methods to detect and quantify genetic ancestry proportions in the data seem adequate for the purpose and I do not see any major problems with the analyses. However, there are some minor issues I wanted to highlight, particularly with regards to the clarity of the statements and the questions set.

1. Abstract ?The BAC groups likely mixed with resident middle-Neolithic farmers (e.g. FBC groups), but not with Neolithic foragers, revealing social stratifications in the Scandinavian Neolithic.? This statement is not exactly in line with the (2d) admixture modelling of BAC section where the YAM+PWC forager model is although less supported than the other but not rejected. So, we cannot rule out such contributions on the basis of the results reported here. Furthermore, I am not sure what exactly could be deduced about the social stratifications. No analyses or statements on this theme are presented in the results section and the discussion ends with a

suggestion that "Future studies will refine our understanding of the social, geographic, and temporal dynamics during this important period in European prehistory." So, it seems the statement in the abstract about social stratifications in the Scandinavian Neolithic is premature and not supported either. I suggest the authors to remove this sentence from the abstract.

2. While the research objectives and questions are implicitly made clear in the introduction there is a certain degree of vagueness throughout the manuscript about these. E.g. (but not limited to these few examples): "We present an archaeogenetic investigation" - in the abstract, and, "we investigate the demographic development in the CWC area", "we paint a detailed picture" in the intro, instead of clearly set up question(s); I suggest the authors to revise the manuscript accordingly. Similarly, the more specific objectives of particular analyses should be presented more clearly: e.g. (2b) "To place the 11 individuals from this study into context" or (2c) "To obtain a clearer picture of the ancestry" - these were probably not really the end goals.

3. Figure 2. Firstly, given that these individual genomes come from a wide geographic area, the authors may consider distinguishing the geography (either by symbols or by colour) on this plot. Secondly, given that the f4 statistic that is shown, $f_4(\text{Chimp}, \text{LBK}; \text{YAM}, \text{X})$, does not explicitly include any hunter-gatherer genomes, "... higher proportion of hunter-gatherer and Neolithic farmer ancestry in CWC and BAC." is not clear as how the higher hunter-gatherer contributions should be read out from the graph. I suggest either to clarify this or remove the word "hunter-gatherer" from this context.

=====

Referee: 2

Malmström et al. 2019 present a simple but insightful archaeogenetic investigation of the demographic origins of individuals associated to the Battle Axe Culture (BAC). By analyzing the genomes the authors reports in their study, together with previously published data, Malmström et al. 2019 show that the BAC represents a group different from earlier populations in Scandinavia. Previous archaeological hypothesis suggested that the BAC complex was a continuation and transformation of the Funnel Beaker Culture (FBC), prehistoric farmers who inhabited southern Sweden.

In addition, this manuscript further builds on demographic dynamics of the arrival of the "steppe-like" ancestry to Neolithic Europe, via Corded Ware Culture (CWC) expansion. Authors find that there is a significant decrease "steppe-like" ancestry with time in CWC/BAC individuals, suggesting that while the initial migration occurred rapidly, later CWC/BAC individuals admixed with local inhabitants of the area.

Overall, the manuscript presented by authors is very well written, thorough in its methodology and sound with its conclusions. Therefore I recommend the publication of this article. However, in order to improve the quality and readability of this manuscript, I suggest the following issues are addressed:

1) Change the "Adimxture Graph" Electronic Supplementary Material (ESM) subsection to "f4 statistic and Adimxture Graph" p.e.

It took me some time to find the one line of text, describing how the f4 analysis was done since it was in a section with a different subheading.

2) In the "Sample preparation" ESM section it mention that "...some libraries were enriched using Caucasian baits and Mybait Human Whole Genome Capture Kit (MYcroarray) following the manufacturer's instructions (Mybaits manual version 2.3.1)."

2.1 I find the term "Caucasian baits" unscientific, uninformative, and unfortunate that company refers to it with such a name. Assuming that the name cannot be modified because of a legal

brand issue, could authors at least quote the term as above to denote the controversy of using that term (i.e. "Caucasian baits")?

2.2 As stated above "some libraries" were enriched using capture enriching techniques. However, it is not specified anywhere in the text, to which individuals does this libraries belong to, how much data was produced from those libraries vs blunt-end ones, nor if they were included in the demographic analyses.

3) Individual's displayed on the x-axis of Fig. 2 represent the point estimates of samples 95% C.I. calibrated dating, an average of the C.I., the upper bound of the C.I. or the lower bound of the C.I.?

In the text where Fig. 2 is being referred, it says "Overall, for CWC associated individuals, there is a clear trend of decreasing affinity to Yamnaya herders with decreasing absolute age (figure 2)." What does "absolute age mean"? Please specify this term, as well as correct the meaning of the dating on the x-axis of Fig. 2, either from Fig. 2 legend or the ESM.

4) The manuscript and ESM makes reference to dates overall with a "BCE" nomenclature, however the archeological description of the Estonian "Karlova" site in the ESM text, as well as Table S1 uses the "BP" nomenclature. Please use the same archaeological dating style consistently. I suggest sticking to the former, and make sure that all instances where an archaeological date is displayed is not using non-updated "BP" dates.

5) Add the geographical/cultural labels from Fig. 1B to Fig. 2

In general, I like the design and display of figures and tables from the paper. However I think that the reader will benefit if authors provided the the geographical/cultural labels from Fig. 1B to the f4 ?decrease of steppe-like ancestry in time? Figure 2, to display the genetic affinities, age and place of origin/culture of investigated individuals in the same figure.

The supplementary table where the ancient DNA (aDNA) reference data is reported (Table S7), doesn't have the individual's dates. While the reader could infer the date of individuals analyzed in Fig. 2, by contrasting the information from Fig. 1B and Fig.2 this is not straight forward.

Thus, in order to more easily convey the results of the paper, as well as to allow the reader to further explore the temporal/geographical distribution of steppe-like ancestry in CWC, I suggest this change is implemented.

6) Yamnaya genetic affinities potentially affected by "data-generated methodological bias" in Fig. 2?

Malmström et al. 2019 test the f4(Chimp, LBK, Yamnaya, X) topology to measure the shared drift between BAC/CWC and Linear Pottery Culture (LBK), as opposed to YAM. Shared genetic drift should increase with higher proportion of hunter-gatherer and Neolithic farmer ancestry in CWC/BAC.

However, this test is potentially affected by using shotgun and captured generated-data in a direct comparisons. Non-captured and captured generated data have different read mapping biases, and directly comparing the shared drift of aDNA data generated with both different strategies can display affinities due to a shared experimental methodology and not to a recent shared demographic history.

According to Table S7 it seems "LBK" individuals are mostly captured-generated (except for the "Stuttgart" sample), while the Yamnaya individuals were both captured and non-captured generated (RISE and I0xx individuals). On one hand, if YAM data was composed mostly of captured-generated data, then the test could be biased against non-captured generated by an artificial affinity signal between the LBK and YAM, which is a conservative scenario.

For instance, there is a "X" sample in Fig. 2 dated around ~ 2900 BCE for which the point estimate and lower SE bound suggests that LBK and YAM are closer to each other, than LBK to "X". In order words, "X" would be an outgroup to both. This is an unexpected demographic scenario,

since if "X" was 100% YAM one should see that LBK is equidistant to both YAM and X. As authors explain, shared drift of "X" and LBK should increase with higher proportion of hunter-gatherer and Neolithic farmer ancestry in CWC/BAC. While the results of this test is not significantly negative, a shift towards negative values suggest an artificial signal is partly influencing this result.

On the other hand, if if YAM data was composed mostly of non-captured data and "X" was captured-generated data one potentially observed a ?exaggerated? affinity between LBK and ?X? just because that data was generated in the same way, although a real demographic signal might exist. In the worst case scenario this could create an artificial decrease of YAM ancestry signal with time in CWC/BAC individuals.

Having said that, "methodological bias" has a higher influence when the data being directly compared is from similar demographic background, which LBK and YAM are not. Nonetheless, in order to improve the readability of the paper and so that future aDNA projects can further understand the effect of ?methodological bias? in direct genetic comparisons, I suggest that authors add the geographical/cultural labels recommended in 5), so that the reader can assess whether captured and non-captured generated data show differences in YAM genetic affinity given the used topology.

Furthermore, I suggest that authors include the discussion above into the suggested ?f4 statistic? section (see #1) of the ESM, so that the reader has more information on the matter. Such section will benefit also from the authors reporting how much data is captured and non-captured from both the LBK and YAM individuals with a small table or described in the text.

Finally, if needed and if there is enough data, authors could repeat the f4 analysis in Fig. 2 using only captured-generated or non-captured generated at the time for both the LBK, YAM groups and "X" tested sample.

Author's Response to Decision Letter for (RSPB-2019-1528.R0)

See Appendix A.

Decision letter (RSPB-2019-1528.R1)

17-Sep-2019

Dear Dr Günther

I am pleased to inform you that your manuscript entitled "The genomic ancestry of the Scandinavian Battle Axe Culture and its relation to the broader Corded Ware horizon" has been accepted for publication in Proceedings B.

Open Access

Your article has been estimated as being 9 pages long. Our Production Office will be able to confirm the exact length at proof stage.

Paper charges

Sincerely,
Victoria Braithwaite

Professor V A Braithwaite
Editor, Proceedings B
<mailto:proceedingsb@royalsociety.org>

Associate Editor, Comments to Author:

I find the authors have done a thorough job of addressing the issues raised.

Appendix A

We thank both reviewers and the associate editor for valuable comments. Below, we have addressed all suggested changes. Below, we have addressed all suggested changes. Our responses to comments are written in blue.

=====

Associate Editor, Comments to Author:

Two reviewers have submitted thoughtful, thorough reviews and both are positive about the subject and about the work. However, both reviewers make a number of suggestions for small but important changes that will make the manuscript clearer and, importantly, more accurate. The authors should pay careful attention to these and justify carefully where any of them have not / cannot be addressed.

=====

Reviewer's Comments to Author:

Referee: 1

This manuscript reports new ancient DNA evidence, in form of whole genome sequence data, from eleven individuals from North and Northeast Europe whose skeletal remains were buried in the context of distinct Middle and Late Neolithic cultures. The main focus of the paper is on Scandinavian Battle Axe Culture (BAC) which culturally belongs to the sphere of the wide-spread Corded Ware Culture (CWC). Previous aDNA studies have shown that burials within the CWC context draw majority of their genetic ancestry from the Steppe Belt region. It has not been clear, however, to what extent BAC represents the spread of CWC to Scandinavia ?as in association with a process of migration or of cultural diffusion and local development?. The genomes of two Swedish BAC individuals analysed in context of the genomes of the earlier Funnel Beaker Culture (FBC) and other Neolithic genomes analysed by the authors and those reported in the literature showed that BAC spread to Scandinavia together with a strong genetic component and with an indication of some level of admixture with FBC, with time series analyses revealing more admixture occurring later on in the Late Neolithic, which is consistent with the trends in Europe reported earlier in the literature. Given the scarcity of genomic data from Neolithic North and Northeast Europe and questions surrounding population origins and continuity I find that this is an important contribution to the field. Of particular interest is the finding of a Megalithic burial with genetic affinity to BAC genomes but without any cultural associations. The methods to detect and quantify genetic ancestry proportions in the data seem adequate for the purpose and I do not see any major problems with the analyses. However, there are some minor issues I wanted to highlight, particularly with regards to the clarity of the statements and the questions set.

1. Abstract ?The BAC groups likely mixed with resident middle-Neolithic farmers (e.g. FBC groups), but not with Neolithic foragers, revealing social stratifications in the Scandinavian Neolithic.? This statement is not exactly in line with the (2d) admixture modelling of BAC section where the YAM+PWC forager model is although less supported than the other but not rejected. So, we cannot rule out such contributions on the basis of the results reported here.

Thank you for this comment, we agree that the statement was not accurate. We have rephrased the sentence in the abstract to 'The BAC groups likely mixed with resident middle-Neolithic farmers (e.g. FBC) without substantial contributions from Neolithic foragers'.

Furthermore, I am not sure what exactly could be deduced about the social stratifications. No analyses or statements on this theme are presented in the results section and the discussion ends with a suggestion that ?Future studies will refine our understanding of the social, geographic, and temporal dynamics during this important period in European prehistory.? So, it seems the statement in the abstract about social stratifications in the Scandinavian Neolithic is premature and not supported either. I suggest the authors to remove this sentence from the abstract.

The reviewer is correct stating that 'social' stratification was an unfortunate choice of words. We removed that term. What we were trying to express in the abstract regarding stratification was that, the genetic differentiation seen among people from the three cultural complexes in Neolithic Scandinavia (BAC, PWC and FBC) indicates some level of social stratification. To highlight this, we modified another sentence to 'By analyzing these genomes together with previously published data, we show that the BAC represents a group different from other Neolithic populations in Scandinavia, revealing stratification among cultural groups.'

2. While the research objectives and questions are implicitly made clear in the introduction there is a

certain degree of vagueness throughout the manuscript about these. E.g. (but not limited to these few examples): "We present an archaeogenetic investigation" - in the abstract, and, "we investigate the demographic development in the CWC area", "we paint a detailed picture" in the intro, instead of clearly set up question(s); I suggest the authors to revise the manuscript accordingly. Similarly, the more specific objectives of particular analyses should be presented more clearly: e.g. (2b) "To place the 11 individuals from this study into context" or (2c) "To obtain a clearer picture of the ancestry" - these were probably not really the end goals.

We agree that some parts of the manuscript can be phrased in a more specific manner. To remove vagueness and be more explicit, we made a number of clarifications, including the following:

In the abstract we changed 'We present an archaeogenetic investigation...' to 'To understand the population dynamics in Neolithic Scandinavia and the Baltic Sea area, we investigate the genomes of ...'.

In the introduction we changed 'In the present paper we investigate the demographic development in the CWC area...' to 'To achieve a better understanding of these population dynamics and the BAC introduction to Scandinavia, we investigate the demographic development in the CWC area...'.

In the introduction we changed '...we paint a detailed picture...' to '...we investigate ancestry and admixture to paint a more detailed picture...'.

In the results section 2b) Exploratory analyses we changed 'To place the 11 individuals from this study into context...' to 'To investigate the genomic ancestry of the 11 individuals from this study in context with other ancient individuals...'.

In the result section 2c) Admixture modelling of CWC we changed 'To obtain a clearer picture of the ancestry...' to 'To investigate the spread of CWC-related people into Scandinavia and obtain a clearer picture of the ancestry proportions found in individuals associated with the CWC and the BAC...'.

3. Figure 2. Firstly, given that these individual genomes come from a wide geographic area, the authors may consider distinguishing the geography (either by symbols or by colour) on this plot.

We agree with both reviewers that information about geographical association would make Figure 2 more informative. We have added this dimension of information by using the same colours as in Figure 1.

Secondly, given that the f_4 statistic that is shown, $f_4(\text{Chimp, LBK; YAM, X})$, does not explicitly include any hunter-gatherer genomes, "... higher proportion of hunter-gatherer and Neolithic farmer ancestry in CWC and BAC." is not clear as how the higher hunter-gatherer contributions should be read out from the graph. I suggest either to clarify this or remove the word "hunter-gatherer" from this context.

The reviewer is correct in stating that we are not explicitly testing hunter-gatherer ancestry in this analysis (apart from the small proportion of HG ancestry in LBK), so we have removed the word "hunter-gatherer" from the sentence.

=====

Referee: 2

Malmström et al. 2019 present a simple but insightful archaeogenetic investigation of the demographic origins of individuals associated to the Battle Axe Culture (BAC). By analyzing the genomes the authors reports in their study, together with previously published data, Malmström et al. 2019 show that the BAC represents a group different from earlier populations in Scandinavia. Previous archaeological hypothesis suggested that the BAC complex was a continuation and transformation of the Funnel Beaker Culture (FBC), prehistoric farmers who inhabited southern Sweden.

In addition, this manuscript further builds on demographic dynamics of the arrival of the "steppe-like" ancestry to Neolithic Europe, via Corded Ware Culture (CWC) expansion. Authors find that there is a significant decrease "steppe-like" ancestry with time in CWC/BAC individuals, suggesting that while the initial migration occurred rapidly, later CWC/BAC individuals admixed with local inhabitants of the area. Overall, the manuscript presented by authors is very well written, thorough in its methodology and sound with its conclusions. Therefore I recommend the publication of this article. However, in order to improve the quality and readability of this manuscript, I suggest the following issues are addressed:

1) Change the "Adimxture Graph" Electronic Supplementary Material (ESM) subsection to "f4 statistic and Adimxture Graph" p.e. It took me some time to find the one line of text, describing how the f_4 analysis was done since it was in a section with a different subheading.

We changed the title of the "Admixture graphs" subsection in ESM to "f4 statistic and Admixture graphs", both in ESM Table of contents and in the subsection heading. Changes were not tracked in the ESM.

2) In the "Sample preparation" ESM section it mention that ..."some libraries were enriched using Caucasian baits and Mybait Human Whole Genome Capture Kit (MYcroarray) following the manufacturer's instructions (Mybaits manual version 2.3.1)."

2.1 I find the term "Caucasian baits" unscientific, uninformative, and unfortunate that company refers to it with such a name. Assuming that the name cannot be modified because of a legal brand issue, could authors at least quote the term as above to denote the controversy of using that term (i.e. "Caucasian baits")?

We agree that the term Caucasian baits is extremely unfortunate. We changed the sentence in ESM subsection Sample preparation from "...enriched using Caucasian baits and Mybait Human Whole Genome Capture Kit (MYcroarray)..." to "...some libraries were enriched using Mybait Human Whole Genome Capture Kit with "Caucasian baits" (both from MYcroarray)..."'. Changes were not tracked in the ESM.

2.2 As stated above "some libraries" were enriched using capture enriching techniques. However, it is not specified anywhere in the text, to which individuals does this libraries belong to, how much data was produced from those libraries vs blunt-end ones, nor if they were included in the demographic analyses.

To clarify this point, we now specify to which individuals the Whole Genome Capture libraries belong, and also that the capture was performed on previously produced blunt-end libraries, by changing the sentence in SEM subsection sample preparation on p. 13 from..."some libraries were enriched..." to "...some libraries (one blunt-end library from ber1, ber2 and ajv54 and two blunt-end libraries from oll009 and oll010) were enriched..."'. The changes were not tracked in the ESM.

The shotgun sequenced Whole Genome Capture data was merged first on library level, together with the blunt-end libraries that were used as templates for the capture. After filtering, all libraries from one individual were merged into one bam file. To clarify this, and to show the approximate proportions of shotgun sequenced Whole Genome capture versus blunt-end data generated, we added the following sentences to ESM subsection Bioinformatic data processing:

'All sequencing runs of the same library, including Whole Genome Captured blunt-end libraries, were merged into one bam file with samtools before duplicate removal. The proportion of human reads deriving from shotgun sequenced Whole Genome Capture data before duplicate filtering was relatively low for the BAC individuals ber1 (2.2%) and ber2 (8.9%) and for the PWC individual ajv54 (14.1%) and higher for the Scandinavian Late Neolithic individuals oll009 (69.9%) and oll010 (89.0%).' We also added... 'All libraries for a single individual were then merged into one bam file using samtools.' and ... 'All data was used for downstream analyses' to the ESM subsection Bioinformatics data processing.'

3) Individual's displayed on the x-axis of Fig. 2 represent the point estimates of samples 95% C.I. calibrated dating, an average of the C.I., the upper bound of the C.I. or the lower bound of the C.I.? In the text where Fig. 2 is being referred, it says "Overall, for CWC associated individuals, there is a clear trend of decreasing affinity to Yamnaya herders with decreasing absolute age (figure 2)." What does "absolute age mean"? Please specify this term, as well as correct the meaning of the dating on the x-axis of Fig. 2, either from Fig. 2 legend or the ESM.

We have rephrased the sentence which now reads "Overall, for CWC associated individuals, there is a clear trend of decreasing affinity to Yamnaya herders with time (figure 2)."

The x-axis in Fig. 2 represents mid-point values of the dates reported in the original publications of the samples. The dates were either calibrated radiocarbon dates, with 95% or 68% CI, or estimated based on archaeological contexts. We could not re-calibrate all dates to 95% CI as some of the original publications did not report the conventional radiocarbon ages BP. However, the mid-points of the time-intervals should not be sensitive to the differences in CI between samples.

We have added this information on the ages to the figure legend of Fig 2. We also added a new table to ESM (table S8) which displays the sample names, sample sites and the dates used to generate Fig. 2.

4) The manuscript and ESM makes reference to dates overall with a "BCE" nomenclature, however the archeological description of the Estonian "Karlova" site in the ESM text, as well as Table S1 uses the "BP" nomenclature. Please use the same archaeological dating style consistently. I suggest sticking to the former, and make sure that all instances where an archaeological date is displayed is not using non-updated "BP" dates.

The "BP" use in the ESM archaeological description of the Estonian "Karlova" individual and in Table S1 refers to the uncalibrated conventional age dates from the radiocarbon laboratories and are reported according to international standard format (e.g. Poz-15499: 3805±35 BP) and can therefore not be changed to BCE.

However, we see the need to clarify this, and have therefore i) modified ESM Table S1 by adding a column with the calibrated BCE values after the uncalibrated BP values and ii) expanded the legend for Table S1 to: 'Conventional radiocarbon ages (i.e. uncalibrated BP values) and calibrated dates (BCE) of ten AMS dated samples in this study and IRMS stable carbon and nitrogen isotope data for eight of the samples.'

Note that for the "Karlova" individual, we do write "The mandible has previously been AMS radiocarbon dated to 2440-2140 cal BCE (95.4%) (Poz-15499: 3805±35 BP, table 1) (16,59)". This means that we first refer to the calibrated radiocarbon date in BCE, with the nomenclature that we use throughout the manuscript; we then add the information (in standard format) on the uncalibrated value as well as the references to where this radiocarbon date was published.

5) Add the geographical/cultural labels from Fig. 1B to Fig. 2

In general, I like the design and display of figures and tables from the paper. However I think that the reader will benefit if authors provided the the geographical/cultural labels from Fig. 1B to the f4 ? decrease of steppe-like ancestry in time? Figure 2, to display the genetic affinities, age and place of origin/culture of investigated individuals in the same figure.

The supplementary table where the ancient DNA (aDNA) reference data is reported (Table S7), doesn't have the individual's dates. While the reader could infer the date of individuals analyzed in Fig. 2, by contrasting the information from Fig. 1B and Fig.2 this is not straight forward. Thus, in order to more easily convey the results of the paper, as well as to allow the reader to further explore the temporal/geographical distribution of steppe-like ancestry in CWC, I suggest this change is implemented.

We agree with both reviewers that information about geographical association would make Figure 2 more informative. We have added this dimension of information by using the same colours as in Figure 1.

As mentioned above, we also added a new table to ESM (Table S8) which displays the sample names, sample sites and the dates used to generate Fig. 2.

6) Yamnaya genetic affinities potentially affected by "data-generated methodological bias" in Fig. 2? Malmström et al. 2019 test the f4(Chimp, LBK, Yamnaya, X) topology to measure the shared drift between BAC/CWC and Linear Pottery Culture (LBK), as opposed to YAM. Shared genetic drift should increase with higher proportion of hunter-gatherer and Neolithic farmer ancestry in CWC/BAC.

However, this test is potentially affected by using shotgun and captured generated-data in a direct comparisons. Non-captured and captured generated data have different read mapping biases, and directly comparing the shared drift of aDNA data generated with both different strategies can display affinities due to a shared experimental methodology and not to a recent shared demographic history.

The reviewer is highlighting an important potential issue in the field: mixing of different data types and potential batch effects. We agree with the reviewer's statement below that this would have "a higher influence when the data being directly compared is from similar demographic background, which LBK and YAM are not", but we still provide more details by adding additional analyses to the ESM (see description below).

According to Table S7 it seems "LBK" individuals are mostly captured-generated (except for the "Stuttgart" sample), while the Yamnaya individuals were both captured and non-captured generated (RISE and I0xx individuals). On one hand, if YAM data was composed mostly of captured-generated data, then the test could be biased against non-captured generated by an artificial affinity signal between the LBK and YAM, which is a conservative scenario.

For instance, there is a "X" sample in Fig. 2 dated around ~ 2900 BCE for which the point estimate and lower SE bound suggests that LBK and YAM are closer to each other, than LBK to "X". In other words, "X" would be an outgroup to both. This is an unexpected demographic scenario, since if "X" was 100% YAM one should see that LBK is equidistant to both YAM and X. As authors explain, shared drift of "X" and LBK should increase with higher proportion of hunter-gatherer and Neolithic farmer ancestry in CWC/BAC. While the results of this test is not significantly negative, a shift towards negative values suggest an artificial signal is partly influencing this result.

We note that this particular individual (ZVEJ28) also falls within the YAM variation on PC2 of Figure 1a, some YAM individuals are even closer to other CWC individuals in that analysis. Furthermore, the same individual is estimated to have ~90% YAM ancestry in Fig 1b, one of the highest proportions of all CWC individuals. In that respect, the position of ZVEJ28 close to YAM is consistent with the other analyses (but this does not fully rule out our potential methodological biases). In our additional analysis (see detailed description below), we see that the f_4 values for this individual are never significantly different from zero regardless of whether SNP capture or shotgun data are used as reference populations.

On the other hand, if if YAM data was composed mostly of non-captured data and "X" was captured-generated data one potentially observed a "exaggerated" affinity between LBK and "X" just because that data was generated in the same way, although a real demographic signal might exist. In the worst case scenario this could create an artificial decrease of YAM ancestry signal with time in CWC/BAC individuals. Having said that, "methodological bias" has a higher influence when the data being directly compared is from similar demographic background, which LBK and YAM are not.

Nonetheless, in order to improve the readability of the paper and so that future aDNA projects can further understand the effect of "methodological bias" in direct genetic comparisons, I suggest that authors add the geographical/cultural labels recommended in 5), so that the reader can assess whether captured and non-captured generated data show differences in YAM genetic affinity given the used topology.

Furthermore, I suggest that authors include the discussion above into the suggested "f4 statistic" section (see #1) of the ESM, so that the reader has more information on the matter. Such section will benefit also from the authors reporting how much data is captured and non-captured from both the LBK and YAM individuals with a small table or described in the text.

Finally, if needed and if there is enough data, authors could repeat the f_4 analysis in Fig. 2 using only captured-generated or non-captured generated at the time for both the LBK, YAM groups and "X" tested sample.

We thank the reviewer for this suggestion. We added a general discussion of this potential issue to the 'f4 statistic and Admixture graphs' section of the ESM. To explicitly test whether our analysis could be affected by this, we added two figures to the ESM (Fig S9 and S10). First, we only use SNP captured data for both reference groups (LBK and YAM), and, second, we only use shotgun sequenced data for the reference groups. In both figures, we also indicate the data type for our test individuals ("X") by different figure shapes and we calculate the regression lines for both data types separately. Reducing the sample sizes for the reference populations led to higher uncertainties in general as indicated by the longer error bars. Nevertheless, we observe that all regression lines show a positive slope indicating that the YAM ancestry reduces over time regardless of what type of data is used for the analysis.